# Evaluation of Sibel's Advanced Neonatal Epidermal (ANNE) wireless continuous physiological monitor in Nairobi, Kenya

Jesse Coleman[1]*, Amy Sarah Ginsburg[2], William Macharia[3], Roseline Ochieng[3], Dorothy Chomba[3], Guohai Zhou[4], Dustin Dunsmuir[5], Shuai Xu[6], J. Mark Ansermino[5]

1 Evaluation of Technologies for Neonates in Africa (ETNA), Nairobi, Kenya, 2 University of Washington, Seattle, Washington, United States of America, 3 Department of Pediatrics, Aga Khan University, Nairobi, Kenya, 4 Center for Clinical Investigation, Brigham and Women's Hospital, Boston, Massachusetts, United States of America, 5 Anesthesiology, Pharmacology & Therapeutics, The University of British Columbia, Vancouver, British Columbia, Canada, 6 Querrey Simpson Institute for Bioelectronics, Department of Biomedical Engineering, McCormick School of Engineering, Department of Dermatology & Department of Pediatrics, Feinberg School of Medicine, Northwestern University, Evanston, Illinois, United States of America

* denots@gmail.com

## Abstract

### Background

Neonatal multiparameter continuous physiological monitoring (MCPM) technologies assist with early detection of preventable and treatable causes of neonatal mortality. Evaluating accuracy of novel MCPM technologies is critical for their appropriate use and adoption.

### Methods

We prospectively compared the accuracy of Sibel's Advanced Neonatal Epidermal (ANNE) technology with Masimo's Rad-97 pulse CO-oximeter with capnography and Spengler's Tempo Easy reference technologies during four evaluation rounds. We compared accuracy of heart rate (HR), respiratory rate (RR), oxygen saturation ($SpO_2$), and skin temperature using Bland-Altman plots and root-mean-square deviation analyses (RMSD). Sibel's ANNE algorithms were optimized between each round. We created Clarke error grids with zones of 20% to aid with clinical interpretation of HR and RR results.

### Results

Between November 2019 and August 2020 we collected 320 hours of data from 84 neonates. In the final round, Sibel's ANNE technology demonstrated a normalized bias of 0% for HR and 3.1% for RR, and a non-normalized bias of -0.3% for $SpO_2$ and 0.2˚C for temperature. The normalized spread between 95% upper and lower limits-of-agreement (LOA) was 4.7% for HR and 29.3% for RR. RMSD for $SpO_2$ was 1.9% and 1.5˚C for temperature. Agreement between Sibel's ANNE technology and the reference technologies met the *a priori*-defined thresholds for 95% spread of LOA and RMSD. Clarke error grids showed that all HR and RR observations were within a 20% difference.

**Data Availability Statement:** To aid others in their research, our full data set is available on the Vivli platform (https://vivli.org/) and can be requested

with the use of the following digital object identifier: https://doi.org/10.25934/PR00007550.

**Funding:** This study was funded by Bill & Melinda Gates Foundation (https://www.gatesfoundation.org/) grant OPP1196617 received by ASG. The sponsor had no role in study design, data collection and analysis, decision to publish or preparation of the manuscript.

**Competing interests:** I have read the journal's policy and the authors of this manuscript have the following competing interests: Shuai Xu is Founder and Chief Executive Officer at Sibel Health; all other authors declare no competing interests. This does not alter our adherence to PLOS ONE policies on sharing data and materials.

## Conclusion

Our findings suggest acceptable agreement between Sibel's ANNE and reference technologies. Clinical effectiveness, feasibility, usability, acceptability, and cost-effectiveness investigations are necessary for large-scale implementation.

## Introduction

Globally, neonatal mortality remains high with over 2.4 million deaths in 2019, the majority in resource-constrained settings [1]. In Sub-Saharan Africa, most neonatal mortality stems from largely preventable and treatable causes of death, including preterm birth, asphyxia, and infectious diseases [2]. Early detection and treatment of these life-threatening conditions using multiparameter continuous physiological monitoring (MCPM) technologies are critical to improving quality of care and averting deaths [3–6]. Currently, MCPM technologies are not commonly available at labor and delivery sites in resource-constrained settings, in part due to the high cost of equipment and lack of trained personnel [7]. The Evaluation of Technologies for Neonates in Africa (ETNA) is an African-based technology-testing platform established to optimize neonatal technologies and improve neonatal health outcomes in resource-constrained settings. ETNA endeavours to understand real-world clinical feasibility, performance, and accuracy of novel technologies. The current study analyzes the clinical accuracy of an investigational MCPM technology compared to verified reference technologies [8].

## Methods

### Study design and procedures

We conducted an iterative prospective study to assess agreement of heart rate (HR), respiratory rate (RR), peripheral oxygen saturation ($SpO_2$) and chest skin surface temperature measurements from Sibel's Advanced Neonatal Epidermal (ANNE) (Sibel Inc., IL, USA), investigational technology with those measurements from reference technologies. We conducted the study at Aga Khan University, Nairobi (AKU-N), a tertiary healthcare facility in Kenya. Sibel's ANNE vital signs monitoring platform includes two neonatal-sized, non-invasive, adhesive skin sensors attached directly to the skin surface that are capable of continuously measuring and recording HR, RR, $SpO_2$, and skin surface temperature (S1 Fig). Up to 30 hours of data is stored locally within the sensor and wirelessly transmitted to a central database supported by customized software. We compared Sibel's ANNE HR, RR, and $SpO_2$ measurements with those from Masimo's Rad-97 pulse CO-oximeter with capnography (Masimo Corporation, USA) technology as reference. RR from the reference technology was measured by capnography using an infant/pediatric nasal cannula to collect the neonate's exhaled carbon dioxide ($CO_2$) levels. We compared Sibel's ANNE temperature measurements with those measured using Spengler's Tempo Easy non-contact infrared thermometer (SPENGLER HOLTEX Group, Aix-en-Provence, France) as reference.

In order to identify agreement thresholds for comparison with Sibel's ANNE technology, we assed functionality and estimated within- and between-neonate variability while verifying Masimo's Rad-97 technology [8]. We ran an initial round of open-label data collection from both Sibel's ANNE technology and the reference technologies to test the accuracy testing methods. In the open-label round, reference data for HR, RR, and $SpO_2$ was shared with Sibel before analysis. These data included 1 hertz (Hz) trends data (including HR, RR, and $SpO_2$

values), the raw plethysmograph waveform, signal quality data, and the capnography $CO_2$ waveform from Masimo's Rad-97 technology. We then conducted three rounds of closed-label testing and analyses. After each subsequent round of data analysis, Sibel was provided with all reference technology datasets in order to provide Sibel an opportunity to improve their detection and measurement algorithms. The study's primary outcome was agreement between the HR, RR, SpO$_2$, and temperature measurements for Sibel's ANNE technology and the reference technologies. We hypothesized that Sibel's ANNE technology would show good agreement within *a priori*-defined thresholds for each vital sign measurement and minimal bias when compared to the reference technologies.

Trained study clinicians recruited, obtained informed consent, and enrolled eligible neonates from the neonatal intensive care unit (NICU), neonatal high dependency unit (NHDU), and postnatal and maternity wards at AKU-N (Table 1). Neonates were simultaneously

**Table 1. Eligibility criteria and study definitions.**

| Eligibility criteria | |
|---|---|
| Inclusion criteria | • Corrected age of < 28 days<br>• Caregivers willing and able to provide informed consent and available for follow-up for the duration of the study |
| Exclusion criteria | • Receiving continuous positive airway pressure or mechanical ventilation<br>• Skin abnormalities in the nasopharynx and/or oropharynx<br>• Contraindication to skin sensor application<br>• Known arrhythmia<br>• Congenital abnormality requiring major surgical intervention<br>• Any medical or psychosocial condition or circumstance that would interfere with study conduct or for which study participation could put the neonate's health at risk |
| Study definitions | |
| Epoch | A 60-second period of time |
| Breath | One cycle of neonate-initiated inhalation and exhalation |
| Breath start | End of a waveform trough (low point) where the carbon dioxide level starts to ascend |
| Respiratory rate (RR) manual counting protocol | A breath was counted if the waveform peak reached either 15 millimeters of mercury (mmHg) or the average peak of the epoch, AND the waveform trough dipped below the average trough of the epoch plus 10 mmHg |
| | · Each plot was counted by two independent reviewers and averaged |
| | · If the difference in the counts was > 5, a third independent reviewer counted the plot |
| | · If the third count was within 5 breaths of either previous count, the average of the two closest counts was used |
| RR median calculation | For each breath in an epoch, a RR was calculated by determining how many breaths would fit in the epoch, and the median of the RR values in an epoch was calculated |
| RR epoch exclusion criteria | RR epoch excluded if: (a) the difference between the epoch count and median RR was > 10; (b) either value was < 15; c) the capnogram contained a digital artifact; or d) if there was lack of inter-reviewer manual count agreement |
| Heart rate (HR) median calculation | Median of the instantaneous HR values in the epoch |
| Adequate signal quality | Sibel's ANNE investigational technology: signal quality score > 0 for the duration of the epoch |
| | Masimo's Rad-97 reference technology: plethysmograph quality index (PO-SQI) threshold > 150 for HR and SpO$_2$, and capnography quality score (CO$_2$-SQI) threshold $\geq$2 for at least 90% of the epoch for RR |
| | Spengler's Tempo Easy: temperature measurement within normal skin temperature range (35.5–37˚C) |

monitored by Sibel's ANNE technology and reference technologies for a minimum of 1 hour with no upper limit for the duration of monitoring. For temperature, trained and experienced study nurses conducted spot checks with the reference technology once every 10 minutes for the first hour and once per hour of participation thereafter. Neonates exited from the study upon discharge from the ward or following caregiver request to discontinue monitoring.

## Data processing and selection

HR, RR, and $SpO_2$ data were collected from Masimo's Rad-97 technology in real-time with a custom Android (Google LLC, Mountain View, USA) application. Temperature data were entered manually into a REDCap data collection application [9]. HR, RR, and $SpO_2$ data were parsed in C (Dennis Ritchie & Bell Labs, USA) to obtain plethysmograph waveform and plethysmograph quality index (PO-SQI) data at 62.5 Hz and capnography waveform data at approximately 20 Hz. Instantaneous HR was obtained from the timing of the PO-SQI, which was calculated by Masimo's Rad-97 technology for each heartbeat. We completed analysis of $CO_2$ waveform data using a breath detection algorithm developed in MATLAB (Math Works, USA) based on adaptive pulse segmentation which has been validated internally and on the CapnoBase database [10] and is accurate to within ±5% for a neonate breathing at 60 breaths/ minute [11]. The breath detection timing allowed for a breath duration calculation. An algorithm calculated the RR median for each epoch (Table 1). Furthermore, the custom MATLAB algorithm also provided a capnography quality index ($CO_2$-SQI) based on capnography features. Values for $SpO_2$ were provided by Masimo's Rad-97 at 1 Hz.

We performed manual RR counting from capnography in the reference technology. Two trained observers independently reviewed plotted capnogram waveforms and counted all breaths within each epoch based on standardized rules. The independent counts were averaged; if the number of breaths counted varied by more than three breaths, a third trained observer also counted the breaths, and the two closest results were averaged.

Measurements of HR, RR, and $SpO_2$ from Sibel's ANNE technology were sampled at between 128 and 512 Hz from the output signal and down-sampled to provide values at 1 Hz. Temperature measurements were conducted once every 10 minutes. To evaluate agreement, we included 60-second HR, RR, and $SpO_2$ epochs, with sufficient signal quality, which were randomly selected (S1 Table). All temperature measurements were included in the analysis.

To calculate sample size for each closed-label round, we estimated that 20 neonates with ten replications each would provide a 95% upper and lower LOA between two methods of +/-0.76 times the standard deviation (SD) of their differences. Tight confidence intervals (CI) require sample sizes of roughly 100 to 200 samples which is generally sufficient for method comparison studies [12].

Mean HR, RR, and $SpO_2$ values for the selected epochs were calculated (Table 2). Manual RR counting was performed for each epoch using capnograms for closed-label rounds one through three (S2 Fig).

## Statistical analysis

To determine the normalized agreement between Sibel's ANNE and reference technologies, we calculated the normalized bias (95% CI) and spread between the 95% limits of agreement (LOA) by dividing the bias and spread between the 95% LOA by the overall mean reference value [13]. Based on Masimo's Rad-97 reference technology verification phase, the acceptable *a priori*-defined spread between the 95% upper and lower LOA of 30%, approximately equivalent to a root-mean-square deviation (RMSD) of 8, was selected for both RR and HR [8]. RMSD was calculated for each vital sign. We selected RMSD thresholds of $\leq$ 3.5% for SpO2

**Table 2. Results from Bland-Altman analysis for Sibel ANNE investigational technology versus reference technology.**

| | Open-label | Closed-label round one | Closed-label round two | Closed-label round three |
|---|---|---|---|---|
| Sibel ANNE HR compared to Masimo Rad-97 HR | | | | |
| Included neonates | 10 | 20 | 20 | 20 |
| Mean Masimo Rad-97 HR | 136.8 | 137.6 | 137.2 | 131.0 |
| Normalized bias (95% CI) | 0.2% (0.0 to 0.5%) | 2.2% (1.5 to 3.1%) | 1.4% (0.8 to 2.0%) | 0% (-2.1 to 1.3%) |
| Normalized spread between 95% LOA (upper and lower 95% LOA) | 4.6% (4.0 to 5.3%) | 23.1% (20.3 to 25.9%) | 16.2% (14.2 to 18.1%) | 4.7% (4.2 to 5.3%) |
| Normalized root-mean-square deviation | 1.2% | 6.3% | 4.3% | 1.2% |
| Sibel ANNE RR compared to manual RR count | | | | |
| Included neonates | No manual RR count | 20 | 20 | 20 |
| Mean manual RR count | | 52.6 | 52.1 | 50.2 |
| Normalized bias (95% CI) | | 1.1% (-2.5 to 2.8%) | -19.9% (-16.0 to -23.8%) | 3.1% (2.0 to 4.1%) |
| Normalized spread between 95% LOA (upper and lower 95% LOA) | | 75.1% (66.0 to 84.2%) | 110.1% (96.8 to 123.4%) | 29.3% (25.7 to 37.8%) |
| Normalized root-mean-square deviation | | 19.1% | 34.4% | 8.0% |
| Sibel ANNE RR compared to Masimo Rad-97 RR median | | | | |
| Included neonates | 9 | 20 | 20 | 20 |
| Mean Masimo Rad-97 RR | 51.5 | 54.8 | 54.7 | 52.5 |
| Normalized bias (95% CI) | -14.3% (-9.2 to -19.3%) | -3.8% (-1.2 to -6.4%) | -23.6% (-19.7 to -27.4%) | -1.5% (-0.7 to -2.2%) |
| Normalized spread between 95% LOA (upper and lower 95% LOA) | 119.8% (102.3 to 137.2%) | 74.4% (65.3 to 83.4%) | 108.57% (95.4 to 121.7%) | 20.6% (18.1 to 23.1%) |
| Normalized root-mean-square deviation | 33.6% | 19.3% | 36.5% | 5.4% |
| Sibel ANNE SpO$_2$ compared to Masimo Rad-97 SpO$_2$ | | | | |
| Included neonates | 7 | 20 | 20 | 20 |
| Mean Masimo Rad-97 SpO$_2$ | 95.2% | 93.9% | 94.4% | 93.9% |
| Bias (95% CI) | 0.7% (0.1 to 1.2%) | 0.1% (-0.3 to 0.5%) | 2.7% (2.2 to 3.1%) | -0.3% (-0.5 to 0%) |
| Spread between 95% LOA (upper and lower 95% LOA) | 9.8% (8 to 11.7%) | 11.3% (9.9 to 12.7%) | 13.9% (12.2 to 15.6%) | 7.7% (6.7 to 8.6%) |
| Root-mean-square deviation | 2.6% | 2.9% | 4.4% | 1.9% |
| Sibel ANNE skin surface temperature compared to Spengler Tempo Easy skin surface temperature | | | | |
| Included neonates | 10 | 20 | 20 | 20 |
| Mean Spengler Tempo Easy skin surface temperature (°C) | 35.8 | 35.9 | 35.9 | 35.6 |
| Bias (95% CI) (°C) | -0.1 (-0.3 to 0.1) | 0.3 (0.1 to 0.4) | 0.5 (0.3 to 0.7) | 0.2 (-0.1 to 0.6) |
| Spread between 95% LOA (upper and lower 95% LOA) (°C) | 2.1 (1.4 to 2.8) | 2.5 (1.9 to 3.0) | 3.2 (2.6 to 3.8) | 5.9 (4.8 to 7.0) |
| Root-mean-square deviation (°C) | 0.5 | 0.7 | 1.0 | 1.5 |

Comparisons include heart rate (HR), respiratory rate (RR), oxygen saturation (SpO$_2$) and skin surface temperature measurements during specific rounds of testing.

and ≤ 1.5°C for temperature, with a spread between the 95% upper and lower LOA of ≤ 4.5°C, based on a review of the literature and internal reference technology testing completed during the verification phase of the study [8]. Clarke error grids were constructed with zones of 20% discrepancy to improve clinical interpretability of RR and HR results.

All analyses were conducted using R (version 3.6.2) with the following packages: readr (version 1.3.1), data.table (version 1.12.8), dplyr (version 0.8.5), stringr (version 1.4.0) and ggplot2 (version 3.3.0).

This study was conducted in accordance with the Guideline for Good Clinical Practice/ International Standards Organization (ISO) 14155 to ensure accurate, reliable, and consistent data collection. The study protocol was approved by Western Institutional Review Board

(20191102), Aga Khan University Nairobi Research Ethics Committee (2019/REC-02), and Kenya Pharmacy and Poisons Board (19/05/02/2019(078)). Written informed consent from each neonate's caregiver was obtained in English or Swahili by trained study staff according to a checklist that included ascertainment of caregiver comprehension.

## Results

Between November 7, 2019 and August 28, 2020, 137 neonates were enrolled and 84 were included for analysis (S3 Fig). Four neonates withdrew before the minimum data collection time. Data from 49 neonates were excluded from analysis due to poor quality data identified in Masimo's Rad-97 reference device (31), insufficient length of recording for comparisons (15), or technology issues (3).

Data from 40 (47.6%) female, 43 (51.2%) male, and one (1.2%) intersex neonates were included in the analysis. Neonates were recruited from the NHDU (69%), postnatal and maternity wards (29.7%), and NICU (1.2%). Median gestational age was 38 (range 25 to 42) weeks. Primary diagnoses upon admission were prematurity (36.9%), healthy post-delivery (32.1%), respiratory distress syndrome (9.5%), jaundice (8.3%), hypoglycemia (6%), low birth weight (3.6%), asphyxia (2.4%), and transient tachypnea (1.2%), all of which were mutually exclusive. We collected 320 hours of data with a median length of recording of 240 (range 29 to 417) minutes per neonate.

In the open-label analysis round, 140 epochs were selected from nine neonates for RR, 153 epochs from 10 neonates for HR, 84 epochs from seven neonates for $SpO_2$, and 28 measurements from 10 neonates for temperature. A total of 81.5% of the data from Sibel's ANNE technology was considered sufficient quality in the open-label round, compared with 75.7% of the data from the reference technology (S1 Table). During each closed-label round, 10 epochs were selected from a minimum of 20 neonates for HR, RR, $SpO_2$, and temperature, resulting in 200 measurement pairs per vital sign per round being included. More data from Sibel's ANNE technology were accepted as being sufficient quality in each of the closed-label rounds, compared with the data from the reference technology (round 1: ANNE = 78.4% vs 63.3%; round 2: ANNE = 56.5% vs 50.1%; round 3: ANNE = 84.0% vs 76.1%). No overlapping epochs were in any of the analysis rounds.

Analysis of the HR data showed a small positive normalized average bias (range 0 to 2.2%) with a normalized spread of LOA meeting or surpassing the *a priori*-defined threshold in each round (Table 2; Fig 1). We observed a decrease in the normalized spread between 95% LOA (16.2 to 4.7%) and RMSD (4.3 to 1.2%) between closed-label rounds two and three. All Sibel's ANNE HR measurements were within 20% of Masimo's Rad-97 values (Fig 2A, region A, Clarke error grid).

RR analyses showed a large variation in average bias across rounds for Sibel's ANNE technology compared to Masimo's Rad-97 for both manually counted RR (range -12.9 to 1.5 breaths/minute) and algorithm-derived RR median (range -13.0 to -0.7 breaths/minute) values (Table 2; Fig 3). The normalized spread between 95% LOA decreased between the second and third closed-label rounds for manual RR count (110.1 to 29.3%) and median RR (110.3 to 20.6%), thereby meeting the *a priori*-defined threshold for both methods of calculating RR. Absolute and normalized spreads of 95% LOA for median RR values were smaller than manually counted RR values in all rounds. All Sibel's ANNE RR measurements were within 20% of Masimo's Rad-97 values (Fig 2B, region A, Clarke error grid).

$SpO_2$ analysis showed minimal change in bias (range -0.3 to 2.7%) for Sibel's ANNE technology compared to Masimo's Rad-97, with the largest change occurring between the second and third closed-label rounds (2.7 to -0.3%; Table 2; Fig 4). The RMSD increased between the

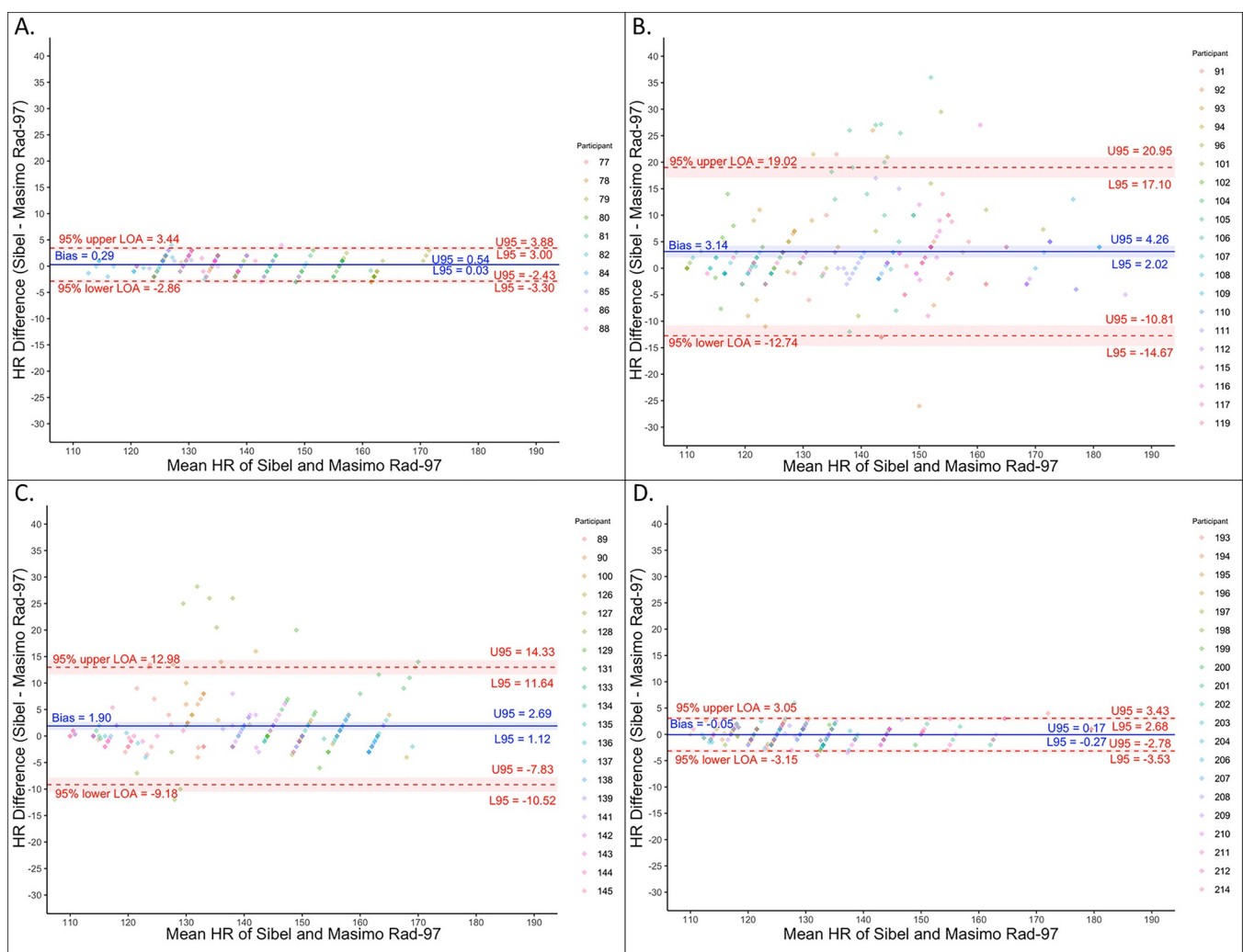

**Fig 1. Bland-Altman plots for heart rate (HR).** (A) Open-label round. (B) Closed-label round one. (C) Closed-label round two. (D) Closed-label round three. Colors indicate which enrolled neonate is associated with the measurement pair.

open-label and second closed-label rounds (2.6 to 4.4%), followed by a decrease to 1.9% between the second and third closed-label rounds, meeting the *a priori*-defined threshold.

Skin surface temperature analysis showed minimal bias and bias change (range -0.1 to 0.5˚C) between rounds for Sibel's ANNE technology compared to Spengler's Tempo Easy reference technology (Table 2; Fig 5). The RMSD for temperature increased (0.5 to 1.5˚C) between each round but met the *a priori*-defined accuracy threshold in each round.

## Discussion

The *a priori*-defined agreement thresholds for neonatal HR, RR, $SpO_2$, and skin surface temperature measurements were met after completing three rounds of closed-label analyses comparing Sibel's ANNE technology and the reference technologies. Between the open and closed rounds, Sibel modified the HR-detection algorithm by adding edge case handlers in the ECG signal where significant motion artifact was detected. Between closed-label rounds two and three, Sibel's ANNE chest sensor software algorithms were augmented to interrogate bio-

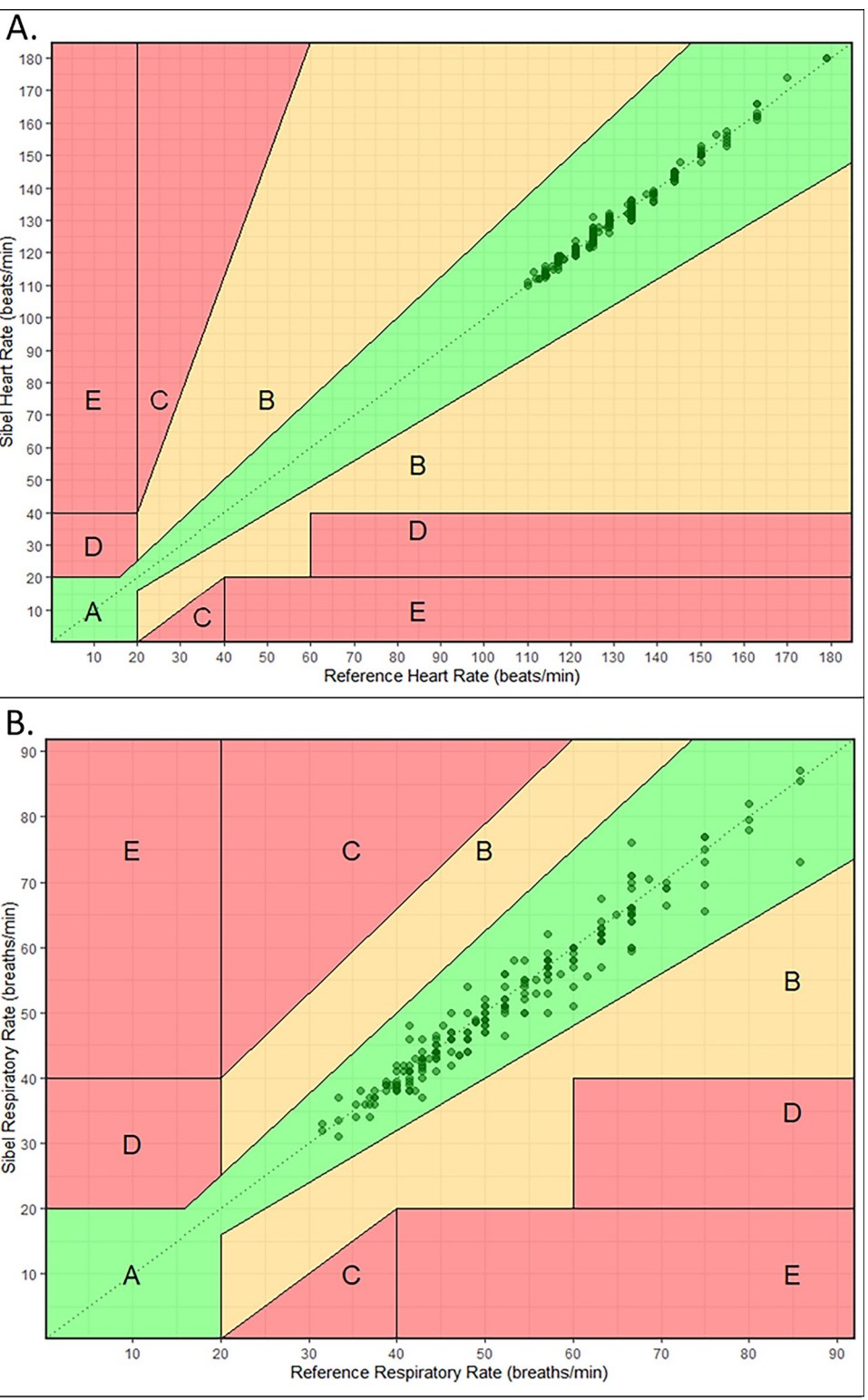

**Fig 2. Clarke error grids for closed-label round three.** (A) Comparison of heart rate (HR) measurements. (B) Comparison of Sibel ANNE respiratory rate (RR) to Masimo Rad-97 RR manual count. Each dot represents a data pair, with the color intensity proportional to density of data pairs. Region A (in green) contains data pairs that are within 20% of the Masimo Rad-97 device value. Region B (in yellow) contains data pairs not within 20% that would not lead to unnecessary treatment. Regions C, D and E are in red. C includes data pairs leading to unnecessary treatment. D includes data pairs with a failure in detecting low or high HR/RR events and E includes data pairs where low and high HR/RR events are confused.

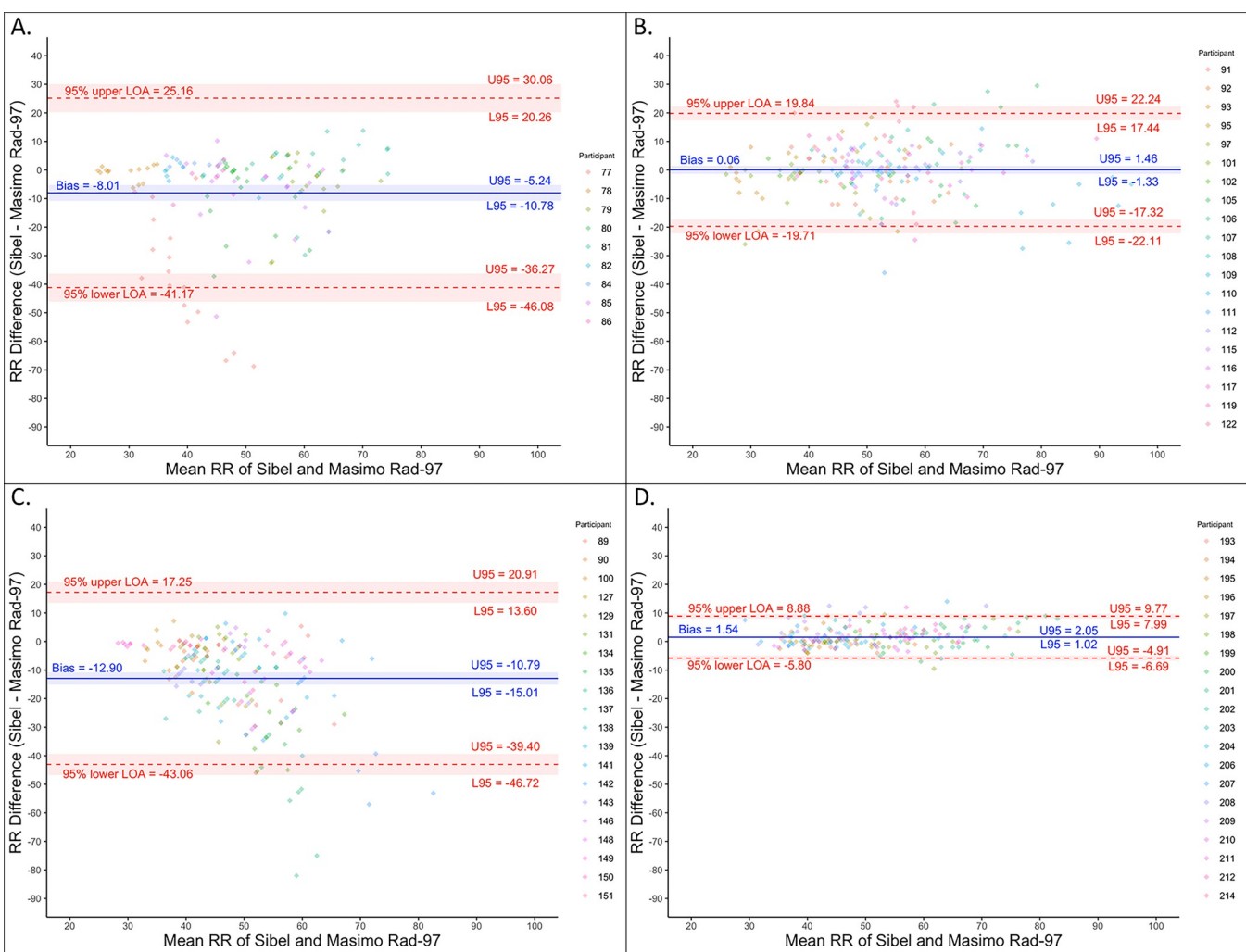

**Fig 3. Bland-Altman plots for Rad-97 respiratory rate (RR) median.** (A) Open-label round. (B) Closed-label round one. (C) Closed-label round two. (D) Closed-label round three. Colors indicate which enrolled neonate is associated with the measurement pair.

impedance measurements for improved RR calculations. A modified calibration factor was also implemented for Sibel's ANNE limb sensor at this stage. Following these modifications, the normalized spreads between 95% LOA for HR, RR, and SpO$_2$ decreased and there was a reduction in bias for all vital signs.

A normalized ±30% spread of 95% LOA for HR and RR was selected using real-world data obtained from neonates during Masimo's Rad-97 reference technology verification phase [8]. A similar LOA has been widely accepted in determining thresholds of agreement for a new method in cardiac output method comparison studies which has been used extensively in the field since it was proposed in 1999 [14]. For a neonate breathing at 60 breaths/minute with a within-neonate variation of 2 breaths/minute, a 30% spread of LOA would equate to 3.3% variation. The Clarke error grids suggest that it is unlikely that treatment decisions would have significantly changed based on the differences between simultaneous observations made by the two technologies.

Capnography has superior performance at higher RR, which is common in neonates, and was chosen as the reference standard for measuring RR [15]. Using a standardized protocol to carefully count breaths from capnograms allowed for manually counted values to be compared

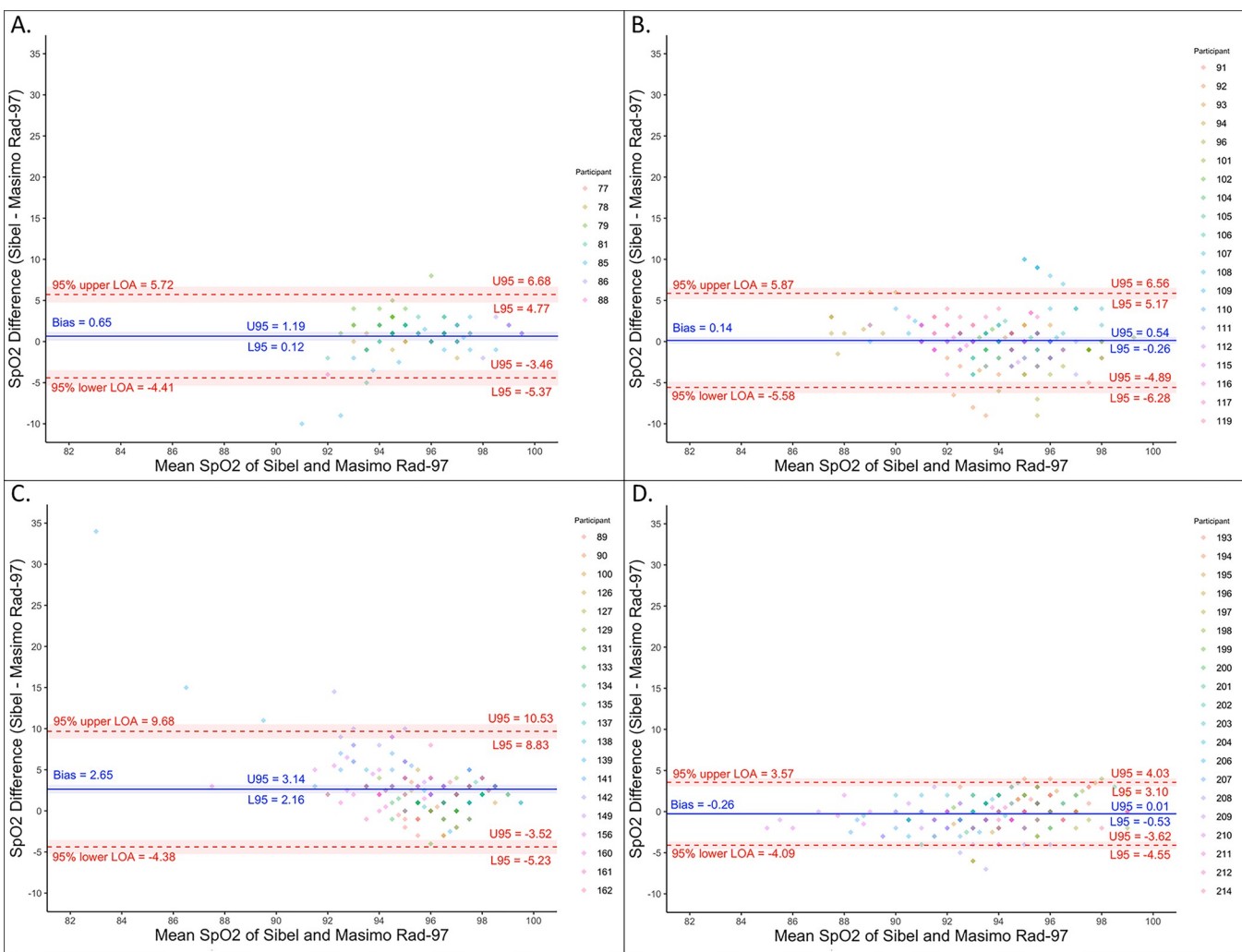

**Fig 4. Bland-Altman plots for oxygen saturation (SpO₂).** (A) Open-label round. (B) Closed-label round one.(C) Closed-label round two. (D) Closed-label round three. Colors indicate which enrolled neonate is associated with the measurement pair.

with the RR values provided by Sibel's ANNE technology. We found that the accuracy of RR comparisons was dependent on the correct placement of Sibel's ANNE sensors. The improved agreement seen in closed-label round three likely was due in part to a change in the chest sensor location from a horizontal placement across the central sternum. In closed-label round three, the chest sensor was placed at a 45-degree angle with one end on the xiphoid process and the other end on the abdomen. This change augmented the signal strength of the bio-impedance signal of RR in neonates, after which RR agreement improved sufficiently to meet the agreement threshold.

Optimizing Sibel's ANNE algorithm between closed-label rounds two and three also resulted in large improvements in SpO₂ accuracy compared to the reference technology. These changes were introduced upon recognizing that the enrolled neonates had darker skin tones than those previously evaluated with Sibel's ANNE technology. The SpO₂ accuracy improved after the photoplethysmography light emission was increased.

Surface thermometers do not reflect core body temperature due to their physical distance from the core [16]. The results from the skin surface temperature comparison showed agreement steadily decreasing between analysis rounds. The large spread in 95% LOA in closed-

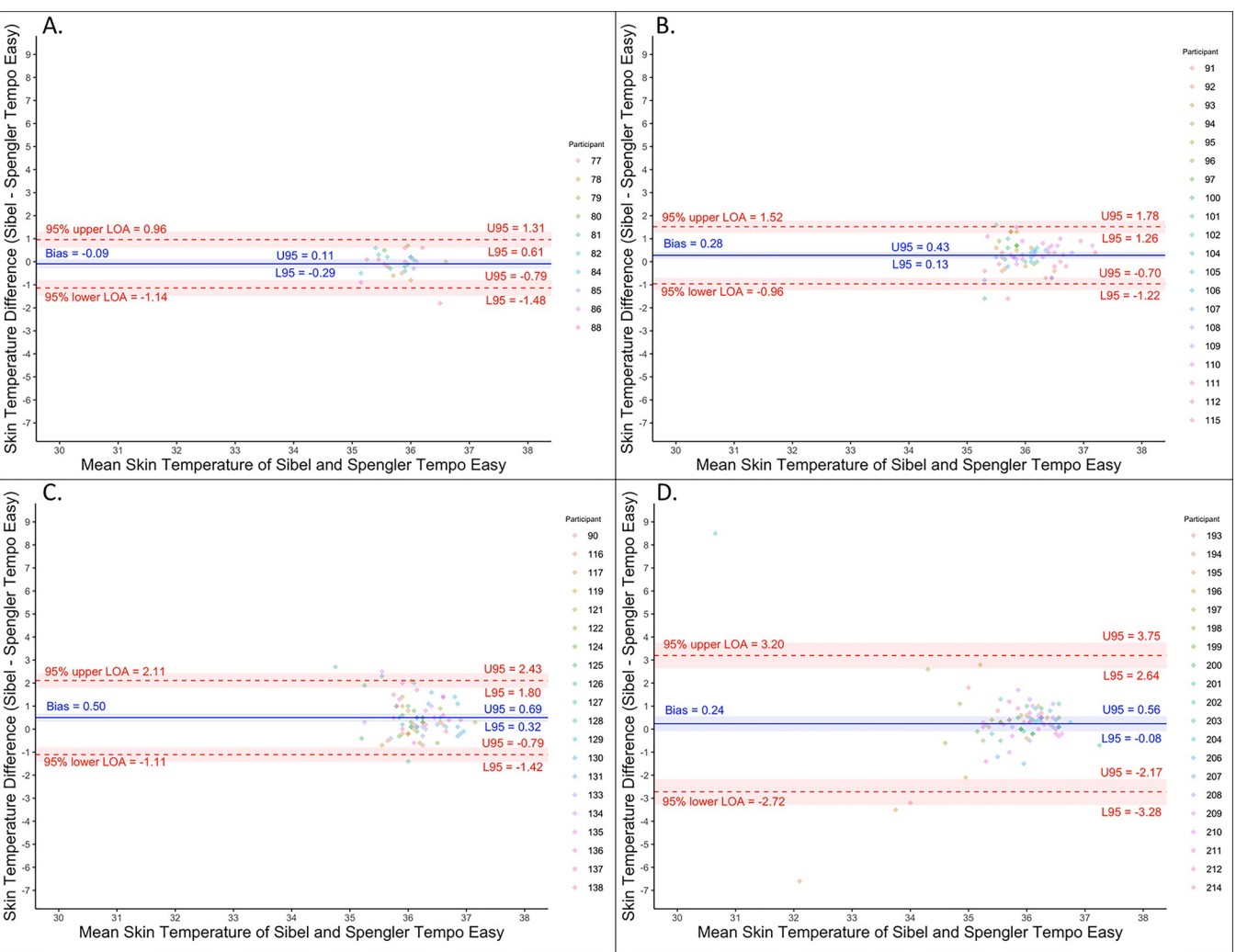

**Fig 5. Bland-Altman plots for skin surface temperature.** (A) Open-label round. (B) Closed-label round one. (C) Closed-label round two. (D) Closed-label round three. Colors indicate which enrolled neonate is associated with the measurement pair.

label round three might be due to three of the 84 (3.6%) temperature values being outside of the 95% upper and lower-LOA by more than 5 degrees. The outlier values may be due to non-compliance with measurement procedures rather than with the accuracy of the technology, but this cannot be verified.

A strength of this study is the non-Sibel investigators' independence in study design, data collection, and analyses. Further, we tested accuracy in the population where the technologies will be used. This led us to discover the impact of darker pigmentation. Our findings are supported by the raw and high-resolution photoplethysmography and capnography data, a manual counting of breaths by two independent reviewers and the randomized selection of comparison epochs. However, the AKU-N study site is relatively highly-resourced and Sibel's ANNE technology may have performed differently in lower-resourced settings. Our recently completed clinical feasibility evaluation of Sibel's ANNE technology at a publicly-funded high-volume maternity hospital is more typical of resource-constrained settings. Usability, acceptability, accuracy, and evaluation of agreement when identifying critical clinical events were also evaluated in this lower-resourced setting.

Sibel's ANNE technology is portable, lightweight, non-invasive and can be battery powered, wireless, and wearable during kangaroo mother care. Its only disposable component is hydrogel adhesive. Of note, data from critically ill neonates with higher or irregular HR, RR, $SpO_2$, or temperature readings could affect Sibel's ANNE sensor performance and impact accuracy comparisons; future accuracy evaluation of Sibel's ANNE technology in neonates in intensive or critical care will be necessary.

## Limitations

There are a number of limitations to the results reported in this study. Approximately one-third (36.8%) of neonate recordings were excluded from the analysis. This was in part due to some fragile neonates not tolerating Masimo's Rad-97 reference technology's nasal cannula. Exclusion due to nasal cannula usage was not a concern with Sibel's ANNE technology because RR is collected from the chest sensor. Electrical outages further affected data quality and duration, contributing to data loss. Furthermore, only epochs with the highest quality reference data were chosen for analysis in order to minimize uncertainty. Bias could have been introduced by the breath detection algorithm during the creation of the capnography quality index ($CO_2$-SQI) which was essential since capnography signal quality was not provided by the reference device. No clinical correlations or outcomes were analyzed as many of the neonates in this study were healthy or relatively healthy.

The accuracy of Sibel's ANNE non-invasive MCPM technology is promising; however, additional research is required prior to large-scale implementation. This could include investigations in clinical care process improvements, clinical outcomes, clinical feasibility, usability, acceptability, cost-effectiveness and clinical effectiveness. The development of a neonatal MCPM suitable for use in resource-constrained settings that can accurately monitor HR, RR, $SpO_2$, and skin surface temperature has promising implications for clinical practice.

## Supporting information

**S1 Fig. A computer rendering of the Sibel Advanced Neonatal Epidermal (ANNE) system investigational vital signs monitoring platform.** The system consists of a chest sensor (L) and a limb sensor (R). The system can respiratory rate, oxygen saturation and skin surface temperature.
(TIF)

**S2 Fig. Masimo Rad-97 reference technology 60-second capnogram demonstrating typical irregularity of respiratory rate.** The monitoring was conducted during a quiet period without external stimuli.
(TIF)

**S3 Fig. Study flow diagram.**
(TIF)

**S1 Table. Overview of Masimo Rad-97 reference and Sibel ANNE investigational technology data from enrolled neonates in the open-label (test) round and subsequent closed-label rounds.**
(TIF)

## Author Contributions

**Conceptualization:** Amy Sarah Ginsburg, William Macharia, Roseline Ochieng, Shuai Xu, J. Mark Ansermino.

**Data curation:** Jesse Coleman, Dorothy Chomba, Guohai Zhou, Dustin Dunsmuir.

**Formal analysis:** Guohai Zhou, Dustin Dunsmuir.

**Funding acquisition:** Amy Sarah Ginsburg, William Macharia, Shuai Xu, J. Mark Ansermino.

**Investigation:** Jesse Coleman, Amy Sarah Ginsburg, Roseline Ochieng, Dorothy Chomba, J. Mark Ansermino.

**Methodology:** Jesse Coleman, Amy Sarah Ginsburg, William Macharia, Dustin Dunsmuir, J. Mark Ansermino.

**Project administration:** Jesse Coleman, Amy Sarah Ginsburg, William Macharia, Shuai Xu, J. Mark Ansermino.

**Resources:** Amy Sarah Ginsburg, William Macharia, Roseline Ochieng, Dorothy Chomba, Shuai Xu.

**Software:** Dustin Dunsmuir, Shuai Xu.

**Supervision:** Jesse Coleman, Amy Sarah Ginsburg, William Macharia, Roseline Ochieng, Shuai Xu, J. Mark Ansermino.

**Validation:** Dustin Dunsmuir.

**Writing – original draft:** Jesse Coleman.

**Writing – review & editing:** Amy Sarah Ginsburg, William Macharia, Roseline Ochieng, Dorothy Chomba, Guohai Zhou, Dustin Dunsmuir, Shuai Xu, J. Mark Ansermino.

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
