## [Decision Letter · Decision Letter 0]

6 Jan 2022

PONE-D-21-28272

Evaluation of Sibel’s Advanced Neonatal Epidermal (ANNE) wireless continuous physiological monitor in Nairobi, Kenya

PLOS ONE

Dear Dr. Coleman,

Thank you for submitting your manuscript to PLOS ONE. After careful consideration, we feel that it has merit but does not fully meet PLOS ONE’s publication criteria as it currently stands. Therefore, we invite you to submit a revised version of the manuscript that addresses the points raised during the review process.

Kindly address the points raised by both reviewers responding to each and providing references to the tracked changes in the revised manuscript. I am sorry it has taken as long as it did. It has been difficult to locate quickly the required number of reviewers during this difficult time.

We look forward to receiving your revised manuscript.

Kind regards,

Martin G Frasch

Academic Editor

PLOS ONE

https://journals.plos.org/plosone/s/file?id=ba62/PLOSOne_formatting_sample_title_authors_affiliations.pdf”

“I have read the journal's policy and the authors of this manuscript have the following competing interests: Shuai Xu is Founder and Chief Executive Officer at Sibel Health; all other authors declare no competing interests.”

Reviewers' comments:

Reviewer's Responses to Questions

**Comments to the Author**

1. Is the manuscript technically sound, and do the data support the conclusions?

Reviewer #1: Yes

Reviewer #2: Partly

2. Has the statistical analysis been performed appropriately and rigorously? 

Reviewer #1: Yes

Reviewer #2: No

3. Have the authors made all data underlying the findings in their manuscript fully available?

Reviewer #1: Yes

Reviewer #2: Yes

4. Is the manuscript presented in an intelligible fashion and written in standard English?

Reviewer #1: Yes

Reviewer #2: Yes

5. Review Comments to the Author

Reviewer #1: The paper presents a comparative study to evaluate the accuracy of a non-invasive neonatal vital signal sensor (Sibel) with measurements available via standard (albeit expensive and inaccessible in many parts of the world) clinical devices. The paper is well-written, and the methods are explained clearly.

One suggestion to improve the paper is to organize the limitations (some of which are now acknowledged by the authors within the text) under a specific heading and/or group them together to give the reader a better sense. In addition to the limitation on exclusion of the data, the rather small number of subjects in the open-label group is an important limitation, in my view.

Reviewer #2: The manuscript reports the results of clinical study to validate the accuracy of the Sibel neonatal monitor against a reference device (Masimo) on a population of neonates in Kenya. The work is important for the adoption of new technology in sub-saharan, and relevant to the scientific community. The study design is sound. The analysis is good, but somehow incomplete as some important aspects are missing, as detailed below.

[DATA PROCESSING & SELECTION]

1. The authors Reference, breath detection algo developed in MATLAB based on ref [10]. The authors should include the expected accuracy of this algorithm, as reported in [10]. Note that, according to [10], that algorithm was only validated on a very small dataset (2 pediatric recordings in the CSL benchmark dataset). I do not expect this to impact the results of this study, but it’s relevant to mention it here so the reader is aware.

2. The analysis is preformed on 60-seconds epochs with sufficient signal quality, randomly selected. The authors should provide more details on how signal quality is assessed, as this could be a possible source of bias in the analysis. Table 1 provides the thresholds applied for Sibel and Masimo to define “sufficient signal quality”. It’s unclear what these SQI mean however, and how the thresholds have been selected.

3. Why did the authors decide to sample epochs from the signals, rather than using all epochs of a pre-defined acceptable quality for the analysis? This would have provided more precision on the LoA estimates. The authors are encouraged to repeat their statistical analysis using the entire data, except maybe for the case of RR (since it requires manual annotations that can be cumbersome on the entire data).

4. It seems like the authors selected epochs independently for the different modalities in the open-label part of the study. Is that so, and if it is, why?

[STATISTICAL ANALYSIS]

5. The acceptable a priori-defined 95% LOA should be specified for SpO2 and Temperature.

6. The acceptable a priori-defined RMSD thresholds for RR and HR should be specified.

7. How were RMSD thresholds set? 4.5C seems very high for temperature, since that could be a difference between a neonate having high fever vs. healthy temperature.

8. Similarly, for RR, the target of 30% LoA spread seems very wide. Reference [12] justifies it by looking at variability between manual and automated annotations of HR and RR. There could be many reasons behind that variability - human error, poor algorithm performance, that are not directly related to the accuracy of the monitoring device studied in this paper. This goes beyond the scope of this manuscript and therefore of this review. I've assumed for this review that the 30% are accepted by the community. But again, this sounds like a very loose performance criterion, and it may be worth adding a comment or remark about it in this paper so that the reader is aware of that assumption and why it's made.

9. The rationale behind the number of epochs and participants (in each branch of the study: open-label, closed-labels round 1-3) is missing. Was it based on LoA precision estimates done on previous data? What was the expected precision the authors were hoping to reach with that sample size? This should be added to the statistical analysis section.

[RESULTS / DISCUSSION]

10. The authors should report the percentage of data that was considered of sufficient quality for both Sibel and the reference systems. It is included in Table S1, but it should be included in the results section as well, e.g. as the percentage of data that was discarded through that process of selecting good quality data. This is an important aspect of device performance as well, next to accuracy when the data is of good quality.

11. In the discussion section, it is stated that “The outlier values are more likely due to non-compliance with measurement procedures than with the accuracy of the technology.” If that’s indeed an issue with non-compliance to measurement procedure, these should be labelled as such, and removed from the analysis as part of the pre-processing and selection process. If it can’t be attributed to a non-compliance issue for sure, then they should be kept in the analysis indeed, and that comment should be rephrased.

12. Figure 1. How do you explain the large increase in LoA when going from open-label to closed-label round #1? The authors explain the reduction in spread between closed-label rounds by a modified calibration factor, but it’s clear why there is such a big jump between the open-label and the closed-label.

6. PLOS authors have the option to publish the peer review history of their article (what does this mean?). If published, this will include your full peer review and any attached files.

Reviewer #1: No

Reviewer #2: No

---

## [Author Response · Author response to Decision Letter 0]

7 Mar 2022

Response to Editorial Board and Reviewers

We thank the editorial board and reviewers for their time and effort in providing valuable feedback and we are grateful to you for the insightful comments. We have incorporated changes to reflect the feedback provided. Our responses to specific points raised are below:

Reviewer #1

Point 1: The paper presents a comparative study to evaluate the accuracy of a non-invasive neonatal vital signal sensor (Sibel) with measurements available via standard (albeit expensive and inaccessible in many parts of the world) clinical devices. The paper is well-written, and the methods are explained clearly.

One suggestion to improve the paper is to organize the limitations (some of which are now acknowledged by the authors within the text) under a specific heading and/or group them together to give the reader a better sense. In addition to the limitation on exclusion of the data, the rather small number of subjects in the open-label group is an important limitation, in my view.

Response:.We thank you for your suggestion. We have moved the discussion of each of these points to a section specific section on limitations that specifically addresses each of these concerns. Furthermore, we had added an explanation as to why the open-label group was necessarily smaller than the closed-label groups.

The updated limitations section, on pages 22-23 reads as follows:

“There are a number of limitations to the results reported in this study. Approximately one-third (36.8%) of neonate recordings were excluded from the analysis. This was in part due to some fragile neonates not tolerating Masimo’s Rad-97 reference technology’s nasal cannula. Exclusion due to nasal cannula usage was not a concern with Sibel’s ANNE technology because RR is collected from the chest sensor. Electrical outages further affected data quality and duration, contributing to data loss. Furthermore, only epochs with the highest quality reference data were chosen for analysis in order to minimize uncertainty. Bias could have been introduced by the breath detection algorithm during the creation of the capnography quality index (CO2-SQI) which was essential since capnography signal quality was not provided by the reference device. No clinical correlations or outcomes were analyzed as many of the neonates in this study were healthy or relatively healthy.”

The updated methods section, which describes the open-label testing round in further detail on pages 5-6 now reads: 

“We ran an initial round of open-label data collection from both Sibel’s ANNE technology and the reference technologies to test the accuracy testing methods. In the open-label round, reference data for HR, RR, and SpO2 was shared with Sibel before analysis.”

Reviewer #2:

The manuscript reports the results of clinical study to validate the accuracy of the Sibel neonatal monitor against a reference device (Masimo) on a population of neonates in Kenya. The work is important for the adoption of new technology in sub-saharan, and relevant to the scientific community. The study design is sound. The analysis is good, but somehow incomplete as some important aspects are missing, as detailed below.

[DATA PROCESSING & SELECTION]

Point 1: The authors Reference, breath detection algo developed in MATLAB based on ref [10]. The authors should include the expected accuracy of this algorithm, as reported in [10]. Note that, according to [10], that algorithm was only validated on a very small dataset (2 pediatric recordings in the CSL benchmark dataset). I do not expect this to impact the results of this study, but it’s relevant to mention it here so the reader is aware.

Response: Thank you for this feedback and we agree with the points raised. In response we have modified the text to include the following text on page 8: 

“We completed analysis of CO2 waveform data using a breath detection algorithm developed in MATLAB (Math Works, USA) based on adaptive pulse segmentation which has been validated internally and on the CapnoBase database [10] and is accurate to within ±5% for a neonate breathing at 60 breaths/minute.[11]”

Point 2: The analysis is preformed on 60-seconds epochs with sufficient signal quality, randomly selected. The authors should provide more details on how signal quality is assessed, as this could be a possible source of bias in the analysis. Table 1 provides the thresholds applied for Sibel and Masimo to define “sufficient signal quality”. It’s unclear what these SQI mean however, and how the thresholds have been selected.

Response: Thank you for that feedback. Monitoring for extended periods of time in a clinical environment will result in artifacts and other disturbances that will corrupt physiological monitoring performance. When comparing devices, it is important to ensure that the same disturbance rejection processes were followed to ensure compatibility of performance. In addition, disturbances may be present in only one device due to the different locations. We used the quality indices developed by the device manufacturers when possible. We did need to develop an SQI for RR for the reference device as this was not provided. We agree and have added that this may have introduced bias. We have added this to the limitations on page 23:

“...bias could have been introduced by the breath detection algorithm when it built the capnography quality index (CO2-SQI) which was essential since capnography signal quality was not provided by the reference device.”

Point 3: Why did the authors decide to sample epochs from the signals, rather than using all epochs of a pre-defined acceptable quality for the analysis? This would have provided more precision on the LoA estimates. The authors are encouraged to repeat their statistical analysis using the entire data, except maybe for the case of RR (since it requires manual annotations that can be cumbersome on the entire data).

Response: We agree that this could have been done. We decided a priori to perform a random sample of epochs based on the literature and our sample size estimation (which is now included). We had also balanced samples across cases with the same number for each case. We agree that the LOA might be reduced if we had a larger sample. However, we performed a validation phase based on the selected sampling procedure to determine the target thresholds. During the initial verification phase, we used many more epochs per case (and provided appropriate correction for this). The closed-label analysis, therefore, had a similar number of epochs but in fewer cases. Your suggested analysis would be interesting but as there were many hundreds more epochs it would take significant effort and the result would not be comparable to the verification phase.

Point 4: It seems like the authors selected epochs independently for the different modalities in the open-label part of the study. Is that so, and if it is, why?

Response: The reason for selecting independent epochs for each variable (modality) is that each variable has its own data file that includes the signal quality index. It would have been extremely complex and onerous to try to find an epoch with all variables (modalities) on both devices with appropriate signal quality all at the exact same time. We hope this addresses this concern as the question was somewhat unclear.

[STATISTICAL ANALYSIS]

Point 5: The acceptable a priori-defined 95% LOA should be specified for SpO2 and Temperature.

Response: Thank you for this feedback. We believe that agreement (with 95% LOA) is a better method for these method comparison studies (see for example Abu-Arafeh A, Jordan H, Drummond G. Reporting of method comparison studies: a review of advice, an assessment of current practice, and specific suggestions for future reports. Br J Anaesth. 2016 Nov;117(5):569-575. doi: 10.1093/bja/aew320. PMID: 27799171.) However, there are ISO standards that still use RMSD, such as ISO 80601-2-61. To allow for comparisons with other devices and studies (especially for SpO2) we chose to focus on RMSD for SpO2 and Temperature. To clarify this issue, we have modified the text on page 10: 

“We selected RMSD thresholds of ≤ 3.5% for SpO2 and ≤ 1.5ºC for temperature, with a spread between the 95% upper and lower LOA of ≤ 4.5ºC, based on a review of the literature and internal reference device testing completed during the verification phase of the study.[14]”

Point 6: The acceptable a priori-defined RMSD thresholds for RR and HR should be specified.

Response: We appreciate this input. We have updated the text to include RMSD for RR and HR on page 10 to read:

“Based on Masimo’s Rad-97 reference technology verification phase, the acceptable a priori-defined spread between the 95% upper and lower LOA of 30%, approximately equivalent to a root-mean-square deviation (RMSD) of 8, was selected for both RR and HR.[14]”

Point 7: How were RMSD thresholds set? 4.5C seems very high for temperature, since that could be a difference between a neonate having high fever vs. healthy temperature.

Response: Thank you for your comment as you have identified a mistake in the manuscript. The RMSD threshold for temperature should read ≤ 1.5ºC. It was the spread between the 95% upper and lower LOA that is ≤ 4.5ºC. To confirm, these RMSD thresholds were selected based on an extensive review of the literature and on the verification phase. To ensure clarity we have corrected the mistake and updated the text on page 10 to read:

“We selected RMSD thresholds of ≤ 3.5% for SpO2 and ≤ 1.5ºC for temperature, with a spread between the 95% upper and lower LOA of ≤ 4.5ºC, based on a review of the literature and internal reference technology testing completed during the verification phase of the study.[14]”

Point 8: Similarly, for RR, the target of 30% LoA spread seems very wide. Reference [12] justifies it by looking at variability between manual and automated annotations of HR and RR. There could be many reasons behind that variability - human error, poor algorithm performance, that are not directly related to the accuracy of the monitoring device studied in this paper. This goes beyond the scope of this manuscript and therefore of this review. I've assumed for this review that the 30% are accepted by the community. But again, this sounds like a very loose performance criterion, and it may be worth adding a comment or remark about it in this paper so that the reader is aware of that assumption and why it's made.

Response: Thank you for this comment. We believe that this is in keeping with other similar comparisons of physiological monitoring such as cardiac output (Critchley LA, Critchley JA. A meta-analysis of studies using bias and precision statistics to compare cardiac output measurement techniques. J Clin Monit Comput. 1999;15: 85–91. doi:10.1023/a:1009982611386). We found a similar spread during the verification phase using two reference devices. This should not be confused with a mean percentage difference. We have added a justification to the text on page 20:

“A similar LOA has been widely accepted in determining thresholds of agreement for a new method in cardiac output method comparison studies which has been used extensively in the field since it was proposed in 1999.[14] For a neonate breathing at 60 breaths/minute with a within-neonate variation of 2 breaths/minute, a 30% spread of LOA would equate to 3.3% variation.”

Point 9: The rationale behind the number of epochs and participants (in each branch of the study: open-label, closed-labels round 1-3) is missing. Was it based on LoA precision estimates done on previous data? What was the expected precision the authors were hoping to reach with that sample size? This should be added to the statistical analysis section.

Response: Thank you for this comment. The precision was calculated before the study was conducted and reported in previously published articles (including Coleman J, Ginsburg AS, Macharia WM et al. Identification of thresholds for accuracy comparisons of heart rate and respiratory rate in neonates [version 2]. Gates Open Res 2021, 5:93) We have added the precision overview to the text on pages 9:

“To calculate sample size for each closed-label round, we estimated that 20 neonates with ten replications each would provide a 95% upper and lower LOA between two methods of +/-0.76 times the standard deviation (SD) of their differences. Tight confidence intervals (CI) require sample sizes of roughly 100-200 samples which is generally sufficient for method comparison studies.[12]”

[RESULTS / DISCUSSION]

Point 10: The authors should report the percentage of data that was considered of sufficient quality for both Sibel and the reference systems. It is included in Table S1, but it should be included in the results section as well, e.g. as the percentage of data that was discarded through that process of selecting good quality data. This is an important aspect of device performance as well, next to accuracy when the data is of good quality.

Response: We agree with this recommendation. The following text has been added to the results section on pages 11 and 12: 

“In the open-label analysis round, 140 epochs were selected from nine neonates for RR, 153 epochs from 10 neonates for HR, 84 epochs from seven neonates for SpO2, and 28 measurements from 10 neonates for temperature. A total of 81.5% of the data from Sibel’s ANNE technology was considered sufficient quality in the open-label round, compared with 75.7% of the data from the reference technology (Table S1). During each closed-label round, 10 epochs were selected from a minimum of 20 neonates for HR, RR, SpO2, and temperature, resulting in 200 measurement pairs per vital sign per round being included. More data from Sibel’s ANNE technology were accepted as being sufficient quality in each of the closed-label rounds, compared with the data from the reference technology (round 1: ANNE = 78.4% vs 63.3%; round 2: ANNE = 56.5% vs 50.1%; round 3: ANNE = 84.0% vs 76.1%). No overlapping epochs were in any of the analysis rounds.”

Point 11: In the discussion section, it is stated that “The outlier values are more likely due to non-compliance with measurement procedures than with the accuracy of the technology.” If that’s indeed an issue with non-compliance to measurement procedure, these should be labelled as such, and removed from the analysis as part of the pre-processing and selection process. If it can’t be attributed to a non-compliance issue for sure, then they should be kept in the analysis indeed, and that comment should be rephrased.

Response: Thank you for highlighting this concerning wording. We have rephrased the text on page 21 to read:

“The outlier values may be due to non-compliance with measurement procedures than with the accuracy of the technology, but this cannot be verified.”

Point 12: Figure 1. How do you explain the large increase in LoA when going from open-label to closed-label round #1? The authors explain the reduction in spread between closed-label rounds by a modified calibration factor, but it’s clear why there is such a big jump between the open-label and the closed-label.

Response: You are correct in identifying the large increase in spread of 95% upper and lower LOA for HR between the open- and closed-label rounds. Sibel used the open-label round to optimize their device performance. To reflect this, we’ve added the following text in page 20:

“Between the open and closed rounds, Sibel modified the HR-detection algorithm by adding edge case handlers in the ECG signal where significant motion artifact was detected.”

---

## [Decision Letter · Decision Letter 1]

1 Apr 2022

Evaluation of Sibel’s Advanced Neonatal Epidermal (ANNE) wireless continuous physiological monitor in Nairobi, Kenya

PONE-D-21-28272R1

Dear Dr. Coleman,

We’re pleased to inform you that your manuscript has been judged scientifically suitable for publication and will be formally accepted for publication once it meets all outstanding technical requirements.

Kind regards,

Martin G Frasch

Academic Editor

PLOS ONE

Additional Editor Comments (optional):

Reviewers' comments:

Reviewer's Responses to Questions

**Comments to the Author**

1. If the authors have adequately addressed your comments raised in a previous round of review and you feel that this manuscript is now acceptable for publication, you may indicate that here to bypass the “Comments to the Author” section, enter your conflict of interest statement in the “Confidential to Editor” section, and submit your "Accept" recommendation.

Reviewer #1: All comments have been addressed

Reviewer #2: All comments have been addressed

2. Is the manuscript technically sound, and do the data support the conclusions?

Reviewer #1: Yes

Reviewer #2: Yes

3. Has the statistical analysis been performed appropriately and rigorously? 

Reviewer #1: Yes

Reviewer #2: Yes

4. Have the authors made all data underlying the findings in their manuscript fully available?

Reviewer #1: Yes

Reviewer #2: No

5. Is the manuscript presented in an intelligible fashion and written in standard English?

Reviewer #1: Yes

Reviewer #2: Yes

6. Review Comments to the Author

Reviewer #1: The paper is organized well. The revision adequately addresses my comments on the earlier version of the manuscript.

Reviewer #2: All comments have satisfactorily been addressed by the authors. Regarding Point 3 of the initial review, the authors explained that it would be hard to redo the analysis with all data points, and provided a justification for their approach. This justification is ok, and the point is considered addressed.

7. PLOS authors have the option to publish the peer review history of their article (what does this mean?). If published, this will include your full peer review and any attached files.

Reviewer #1: **Yes: **Soheil Ghiasi

Reviewer #2: No

---

## [Editor Report · Acceptance letter]

20 Jun 2022

PONE-D-21-28272R1 

Evaluation of Sibel’s Advanced Neonatal Epidermal (ANNE) wireless continuous physiological monitor in Nairobi, Kenya 

Dear Dr. Coleman:

I'm pleased to inform you that your manuscript has been deemed suitable for publication in PLOS ONE. Congratulations! Your manuscript is now with our production department. 

Kind regards, 

on behalf of

Dr. Martin G Frasch 

Section Editor

PLOS ONE